# Identifying the Biological Characteristics of Anthracnose Pathogens of Blueberries (*Vaccinium corymbosum* L.) in China

Wei-Kun Feng [1,2], Chong-He Wang [1,2], Yun-Wei Ju [1,2,*], Zeng-Xin Chen [1,2], Xue Wu [1,2] and Dong-Lu Fang [1,2]

1   College of Forestry and Grassland, Nanjing Forestry University, Nanjing 210037, China; 3210100070@njfu.edu.cn (W.-K.F.); 3220100083@njfu.edu.cn (C.-H.W.); 3230100076@njfu.edu.cn (Z.-X.C.); 8230110087@njfu.edu.cn (X.W.); fangdonglu@njfu.edu.cn (D.-L.F.)
2   Co-Innovation Center for Sustainable Forestry in Southern China, Nanjing Forestry University, Nanjing 210037, China
*   Correspondence: jyw6808@njfu.edu.cn

**Abstract:** *Vaccinium corymbosum* L., commonly known as blueberry, is a valuable small fruit tree in terms of its economic significance and is primarily found in China within the regions of Heilongjiang and Jilin provinces. Additionally, it thrives in the areas spanning the middle and lower reaches of the Yangtze River. Beyond their economic value, blueberries play a crucial role within forest ecosystems, serving as a significant energy source and source of nutrients. Their presence contributes significantly to fostering stability, enhancing biodiversity, and acting as an indicator of environmental quality within forest ecosystems. Since August 2022, an unknown leaf disease has been found on a large scale in blueberry fields in Nanjing, Jiangsu Province, China. The disease causes leaf curling, wilting, and even early defoliation, severely reducing the yield and production value of blueberries. The pathogenicity test confirmed the virulence of the isolates (NG5-1, NG5-2, NG5-3, NG5-4, N2-1, and N2-2) against *V. corymbosum*. The two pathogens were identified as *Colletotrichum fructicola* and *C. aenigma* by observing the morphological characteristics of the isolates and combined with multi-locus phylogenetic analyses (ITS, *CAL*, *ACT*, *TUB2*, *ApMat*, and *GAPDH*). Blueberry anthracnose, caused by *C. aenigma*, is the first report of this disease in China. The biological characteristics of *C. aenigma* were investigated under different conditions, including temperature, pH, light conditions, culture medium, and carbon and nitrogen sources. The optimal temperature for growth was determined to be within the range of 25–30 °C; *C. aenigma* exhibits optimal growth at a pH of 7–8. Mycelial growth is favored under conditions of partial light, whereas complete darkness promotes spore production. It was found that PDA medium was the most favorable medium for *C. aenigma* mycelial growth, and MM medium was the best medium for spore production; the most suitable carbon sources for colony growth and spore production were sorbitol and glucose, respectively, and the most suitable nitrogen source was peptone. This study furnishes a theoretical foundation for a more scientifically informed approach to the prevention and control of anthracnose on *V. corymbosum*.

**Keywords:** blueberry; anthracnose; identification; phylogenetic analysis; biological characterization

## 1. Introduction

*Vaccinium corymbosum* L., a member of the genus *Vaccinium* in the Ericaceae family, originates from North America and thrives in diverse habitats like forests, marshes, and grasslands [1]. In forests, *V. corymbosum* predominantly grows in the understory, benefiting from the filtered sunlight they receive and finding protection from strong winds due to the presence of taller trees. Their preference for acidic soils often leads to their presence in coniferous forests [1,2]. In China, blueberries are found mainly in the forested areas of the Daxinganling and Xiaoxinganling mountains, as well as in the middle and lower reaches of the Yangtze River and in the south [3]. Within forest ecosystems, blueberries serve as a vital food source for various wildlife species, such as birds, bears, and small mammals.

Simultaneously, they play a crucial role in regulating soil acidity and nutrient levels, thereby enhancing the overall quality of forest soils. This contribution aids in water conservation and sustains the stability of both soil and water resources within these ecosystems [2,4,5].

*V. corymbosum*, a perennial deciduous berry, is known for its unique taste and high concentration of antioxidants, anthocyanins, tannins, and polysaccharides [6]. These components are thought to prevent and delay the aging process. Blueberry stems and leaves also contain compounds that regulate blood glucose and lipid levels and prevent the onset of neurological aging. This makes them popular among consumers [7]. In recent years, the planting area of blueberries in China has expanded, and the diseases it suffers have caught more public attention. Examples include stem blight caused by *Neofusicoccum vaccinii* [8], anthracnose caused by *Colletotrichum acutatum* and *C. gloeosporioides* [9,10], anthrax caused by *Pestalotiopsis* spp. [11], *Nigrospora oryzae* [12], *Diaporthe phoenicicola* [13], and leaf spot disease caused by *Calonectria pseudoreteaudii* [14]. The diseases have resulted in significant economic losses to the blueberry industry.

*Colletotrichum* spp. is widely spread globally as epiphytes, parasitoids, or endophytes and is among the most significant fungal pathogens of a broad range of plants [15,16]. It has been reported that there are globally 16 species of *Colletotrichum* spp. complex [17–20], which can infect many cash crops, fruit trees, vegetables, and ornamental plants, causing symptoms such as wilting, leaf lesions, and fruit rot, resulting in severe economic losses [21–24], and some species can even directly affect human health; for example, *C. chlorophyti* and *C. gloeosporioides* are capable of causing fungal inflammation of the cornea [25,26]. *Colletotrichum* spp. has a wide variety of species, and their biological characteristics vary from host to host, so accurate identification of these species is crucial in the control process [27].

*Colletotrichum* spp. infects the leaves, stems, fruits, and other organs of blueberries to cause anthracnose, which leads to leaf abscission, shoot wilting, and fruit rot, severely affecting the growth of blueberries and is one of the important factors limiting the yield of blueberries [28–31]. Due to the high similarity of the characteristics of *Colletotrichum* species, classification is very limited by relying solely on morphological features [32,33]. The application and development of molecular biology techniques have led to the use of morphological and multigene systematic analyses as a more dependable means of anthrax fungi classification [34,35]. The internal transcribed space (ITS), calmodulin (*CAL*), actin (*ACT*), beta-tubulin (*TUB2*), Apn2-Mat1-2 intergenic spacer and partial mating type (*ApMat*), chitin synthase (*CHS-1*), and glyceraldehyde-3-phosphate dehydrogenase (*GAPDH*) loci have been used to distinguish various *Colletotrichum* species [27,35–37].

During a survey of blueberry plantations in Nanjing, Jiangsu Province, China, in August 2022, our team discovered reddish-brown spots on the leaf tips or margins of blueberries. The condition had an incidence rate of over 62%, and as it developed, the leaves curled and wilted, and some plants even succumbed to the disease. The objective of this study was to determine the pathogenicity of representative isolates from blueberries and identify the causal species of blueberry anthracnose using morphological characters combined with multilocus phylogenetic analyses.

## 2. Materials and Methods

### 2.1. Sample Collection and Fungal Isolation

In August 2022, at the Blueberry Industrial Base of the National Agricultural High-tech Industrial Demonstration Zone in Nanjing, Jiangsu Province, China, we randomly inspected and recorded the health of blueberry leaves within the collection area (20 m$^2$ in size, 3 m spacing), totaling 200 leaves. The disease status within the collection area was depicted through the ratio of susceptible leaves to the total leaf count, calculated as the number of susceptible leaves divided by the total number of leaves. Six plants were randomly selected, and five leaves with similar symptoms were collected from each of the thirty leaves. The leaves were rinsed with distilled water and dried. Tissue pieces (4 × 4 mm) were cut at the juncture of diseased and healthy areas, surface disinfected with 75% ethanol solution for 30 s, 5% sodium hypochlorite for 3 min, followed by aseptic water

rinsing for 3 times, then dried with sterilized filter paper, inoculated to PDA medium for 5 d of dark cultivation at 25 °C [38], and transferring colony edge hyphae onto a fresh PDA medium to isolate pure cultures [39]. Colonies were grouped according to colony morphology, and all isolates were observed microscopically to select representative isolates.

## 2.2. Pathogenicity Tests

Subsequent experiments were carried out on detached and living leaves to confirm the pathogenicity of the isolates. The collected healthy leaves were wiped with 75% ethanol and rinsed repeatedly in sterile water. After drying on sterile filter paper, the surface was wound with a sterilized needle. The 5 mm plugs were taken from the edge of the isolates, and the mycelial surface was attached to the wound. PDA plugs were used as a control, and three representative isolates from each of the isolates with different colony morphologies were inoculated with five leaves each. A total of 45 leaves were inoculated. The inoculated leaves were placed in sterile Petri dishes containing moistened filter paper and were incubated at $25 \pm 2$ °C for 7 days. Leaf onset was observed every day.

Based on the pathogenicity of isolated leaves, only isolates with typical *Colletotrichum* spp. characteristics were found to be pathogenic. Three representative isolates (NG5-1, NG5-4, and N2-1) were selected from the pathogenic isolates for further pathogenicity testing, thus confirming Koch's postulates. Seedling leaves were wounded by pricking with a sterile needle, and 10 μL of conidia suspensions ($10^6$ conidia/mL) from the isolates were dropped separately onto the wounded site, and leaves treated with the same volume of sterilized $H_2O$ were used as controls [40]. Five leaves were inoculated for each treatment and observed for seven consecutive days, with photographs taken on the third and seventh days to record leaf condition. The leaves that had been inoculated were covered with plastic film, observed once a day, and sprayed with sterile water to maintain humidity. Then, all seedlings were incubated in a greenhouse at a temperature of $25 \pm 2$ °C. The pathogenic fungi were re-isolated from leaves showing typical symptoms and identified via morphological characterization and ITS sequence analysis. Four 3-year-old healthy *V. corymbosum* seedlings without any chemical treatment were used for inoculation in this experiment. The plants used in the experiment were grown in pots filled with sterilized nutrient soil and were obtained from the Blueberry Demonstration Garden of Nanjing National Agricultural High-altitude Zone, Nanjing, China.

## 2.3. DNA Extraction and PCR Amplification

The genomic DNA of fungi was extracted using a modified CTAB method [41]. The following primers were selected for PCR: ITS1/ITS4 [42], ACT512/ACT783 [43], CL1C/CL2C [44], Bt2a/Bt2b [45], AmF/AmR [46], and GDF/GDR [47]. The specific primers were used to analyze the following gene fragments: internal transcribed spacer (ITS), actin (*ACT*), calmodulin (*CAL*), beta-tubulin (*TUB2*), Apn2-Mat1-2 intergenic spacer and partial mating type (*ApMat*), and glyceraldehyde-3-phosphate dehydrogenase (*GAPDH*). The PCR reaction systems were all 25 μL, including 2 × Taq PCR MasterMix (11 μL), DNA (1 μL), each primer (1 μL), and double-deionized $H_2O$ (11 μL). The conditions for the primer reaction are presented in Table 1. PCR amplification products were detected via 1.5% agarose gel electrophoresis and sequenced by Sangon Biotech (Shanghai) Co., Ltd., Nanjing, China.

**Table 1.** PCR reaction conditions.

| Gene | Primer (Forward/Reverse) | PCR: Thermal Cycles: (Annealing Temp. in Bold) |
|---|---|---|
| ITS | ITS1/ITS4 | 94 °C: 3 min, (94 °C: 30 s, **55 °C**: 30 s, 72 °C: 45 s) × 33 cycles, 72 °C: 10 min |
| *CAL* | CL1C/CL2C | 95 °C: 3 min, (95 °C: 30 s, **55 °C**: 30 s, 72 °C: 30 s) × 35 cycles, 72 °C: 10 min |
| *ACT* | ACT512/ACT783 | 94 °C: 3 min, (94 °C: 30 s, **58 °C**: 30 s, 72 °C: 45 s) × 35 cycles, 72 °C: 10 min |
| *TUB2* | Bt2a/Bt2b | 95 °C: 3 min, (95 °C: 30 s, **55 °C**: 30 s, 72 °C: 30 s) × 35 cycles, 72 °C: 10 min |
| *ApMat* | AmF/AmR | 94 °C: 3 min, (94 °C: 1 min, **55 °C**: 30 s, 72 °C: 1 min) × 35 cycles, 72 °C: 10 min |
| *GAPDH* | GDF/GDR | 94 °C: 3 min, (94 °C: 30 s, **58 °C**: 30 s, 72 °C: 45 s) × 35 cycles, 72 °C: 10 min |

### 2.4. Sequence Alignment and Phylogenetic Analyses

Sequence results were aligned, and homology was analyzed using BLAST in the NCBI database. The ITS, *CAL*, *ACT*, *TUB2*, *ApMat*, and *GAPDH* sequences used for the phylogenetic analyses in this study are listed in Table S1. *C. boninense* (CBS 123755) was used as an outgroup. In PhyloSuite V1.2.2, the six sequences were individually aligned to the locus-based reference sequences using MAFFT V7.313's "--auto" strategy and normal alignment mode and then manually edited using BioEdit ver 7.0 [48]. Phylosuite V1.2.2 was used to concatenate the six sequences [49]. For the multilocus phylogenetic analyses, ModelFinder was used to select the best-fit models [50]. Subsequently, IQtree V 1.6.8 and MrBayes V 3.2.6 were used for maximum likelihood (ML) and Bayesian inference (BI), respectively [51,52]. The ML analysis under the edge-linked partition model for 1000 standard bootstraps. [53]. Using a partition model for BI analysis (2 parallel runs, 2,000,000 generations), the top 25% of the sampled data is discarded. Phylogenetic trees were opened and adjusted with the iTOL online website.

### 2.5. Morphological Identification

Three representative pathogenic strains (NG5-1, NG5-4, and N2-1), initially identified as *Colletotrichum* spp. by preliminary observations of colony morphology and microscopy, were inoculated into the center of the PDA medium, and cultures were incubated for 5 days at 25 °C under a 12/12 h light/dark cycle, and colony color, texture, and hyphal morphology were observed and recorded. The appendages of these isolates were cultured from the conidia with a slide culture [20]. Conidia and appressoria were observed, described, and measured using a Mshot ML31 biological microscope (Guangzhou Micro-shot Optical Technology Co., Ltd., Guangzhou, China) (n = 30).

### 2.6. Biological Characteristic

This is the first discovery of *C. aenigma*-causing blueberry anthracnose in China. To determine the colony growth and spore production of *C. aenigma* under different temperatures (5–35 °C, 5 °C as an interval), pH levels (4, 5, 6, 7, 8, 9, and 10), and light conditions (24 h light, 24 h dark, and 12 h light/dark). Mycelial discs (5 mm diameter) were transferred to the center of fresh PDA medium plates of 90 mm diameter and cultured as above. Colony diameters were measured for 7 consecutive days under different conditions. Ten mycelial plugs (5 mm in diameter) were taken from the edge of the colony in each of the above culture conditions and transferred to 100 mL of sterile water at 180 r/min and 25 °C to induce conidia production. After seven days, the blood count plate was used to determine the sporulation of the colonies for each culture condition. Each treatment was replicated three times.

To determine the colony growth and spore production of *C. aenigma* under different culture media, mycelial plugs (5 mm in diameter) of *C. aenigma* were placed in the center of Potato Dextrose Agar Medium (PDA), Czapek Dox Agar (CZA), Minimum Methanol Medium (MM), and Complete Medium (CM), respectively. Cultivation was conducted at 25 °C with a 12 h light/dark cycle. The compositions of the four media are listed in Table 2. Colony diameters were measured for 7 consecutive days under the above conditions. Ten mycelial plugs (5 mm in diameter) were taken from the edge of the colony in each of the above culture conditions and transferred to 100 mL of sterile water at 180 r/min and 25 °C to induce conidia production. After seven days, the blood count plate was used to determine the sporulation of the colonies for each culture condition. Each treatment was repeated three times.

To determine the colony growth and spore production of *C. aenigma* under carbon and nitrogen sources, CZA was used as a basal medium. As a carbon source, sucrose was replaced by equal amounts of glucose, maltose, soluble starch, lactose, fructose, mannitol, and sorbitol. Similarly, peptone, glycine, yeast powder, lysine, ammonium sulfate, urea, and beef paste were used as nitrogen sources in equal amounts instead of $NaNO_3$. Mycelial plugs (5 mm in diameter) were transferred to the center of dishes containing varying carbon

or nitrogen sources. Colony diameters were measured for 7 consecutive days under the above conditions. Ten mycelial plugs (5 mm in diameter) were taken from the edge of the colony of each of the above culture conditions and transferred to 100 mL of sterile water at 180 r/min and 25 °C to induce conidia production. After seven days, the blood count plate was used to determine the sporulation of the colonies for each culture condition. Each treatment was repeated three times.

**Table 2.** Media and ingredients.

| Medium | Ingredients |
|---|---|
| Potato Dextrose Agar | Potato, 200 g; dextrose, 20 g; distilled water, 1 L |
| Czapek's Agar | NaNO$_3$, 3 g; KCl, 0.5 g; KH$_2$PO$_4$, 1 g; MgSO$_4$·7H$_2$O, 0.5 g; FeSO$_4$·7H$_2$O, 0.01 g; sucrose, 20 g; distilled water, 1 L |
| Minimal Medium | NaNO$_3$, 6 g; KCl, 0.52 g; KH$_2$PO$_4$, 1.52 g; MgSO$_4$·7H2O, 0.64 g; Vogel-Bonner salts, 0.01 g; trace element, 1 mL; agar powder, 15 g |
| Complete Medium | 20 × nitrogen salt, 50 mL; trace element, 1 mL; peptone, 2 g; yeast extract, 1 g; casamino acid, 1 g; vitamin, 1 mL; dextrose, 10 g; agar powder, 15 g |

*2.7. Statistical Analysis*

SPSS 21.0 software was used for one-way analysis of variance (ANOVA), and Duncan's test was used for comparison ($p < 0.05$). GraphPad Prism 8.0.2 software was used for plotting.

**3. Results**

*3.1. Field Symptoms and Isolation of Fungi*

In the blueberry industry base of Nanjing National Agricultural Hi-tech Zone, 62% (124/200) of blueberry plants showed leaf spot disease from July to October 2022 in Jiangsu Province, China. Symptoms usually start at the leaf margin or tip, with spots measuring 7–10 mm in diameter and brown necrotic in the center (Figure 1A,B). The spots gradually expand and merge to form a large irregular necrotic spot, resulting in inward curling of the leaves and defoliation. The center of the spot was gray, surrounded by a dark brown margin (Figure 1C,D). A total of 85 fungal colonies were isolated from the diseased leaves. Based on ITS sequences and colony morphology analysis, three distinct colony types—Colletotrichum, Alternaria, and Pestalotiopsis—were identified, with isolation rates of 81% (69/85), 12% (10/85), and 7% (6/85), respectively.

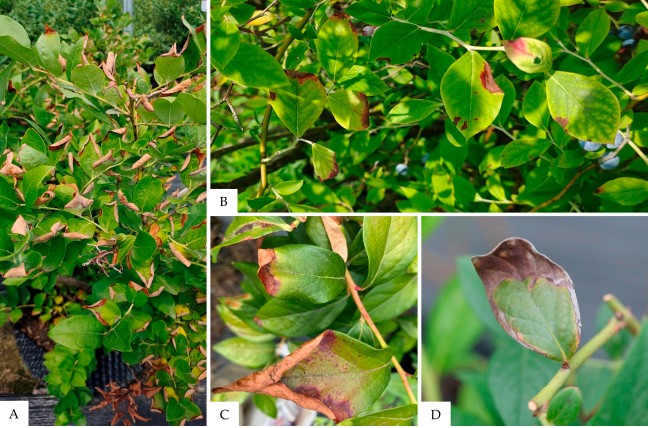

**Figure 1.** Manifestations of leaf blotch under field conditions are depicted in the following manner: (**A**,**B**) illustration of diseased leaves within the field and (**C**,**D**) detailed close-ups of afflicted leaves.

### 3.2. Pathogenicity Test

Six *Colletotrichum* spp. isolates (NG5-1, NG5-2, NG5-3, NG5-4, N2-1, and N2-2) used for inoculation on detached leaves were all found to be pathogenic to blueberries. In contrast, leaves inoculated with *Alternaria* sp., *Pestalotiopsis* sp., and control leaves showed no symptoms. Subsequently, three representative isolates of *Colletotrichum* spp. (NG5-1, NG5-4, and N2-1) were selected for inoculation on healthy attached blueberry seedlings. Three days after inoculation, brown spots appeared on the leaves (Figure 2A–C). The spots seven days after inoculation were significantly enlarged (Figure 2E–G), consistent with symptoms under natural conditions, while control plants remained healthy with no visible symptoms (Figure 2D,H). Fungi reisolated from diseased spots were consistent with the inoculated fungi and shared the same morphological characteristics. The three isolates of *Colletotrichum* spp. were shown to be the pathogenic fungi responsible for blueberry leaf spot disease.

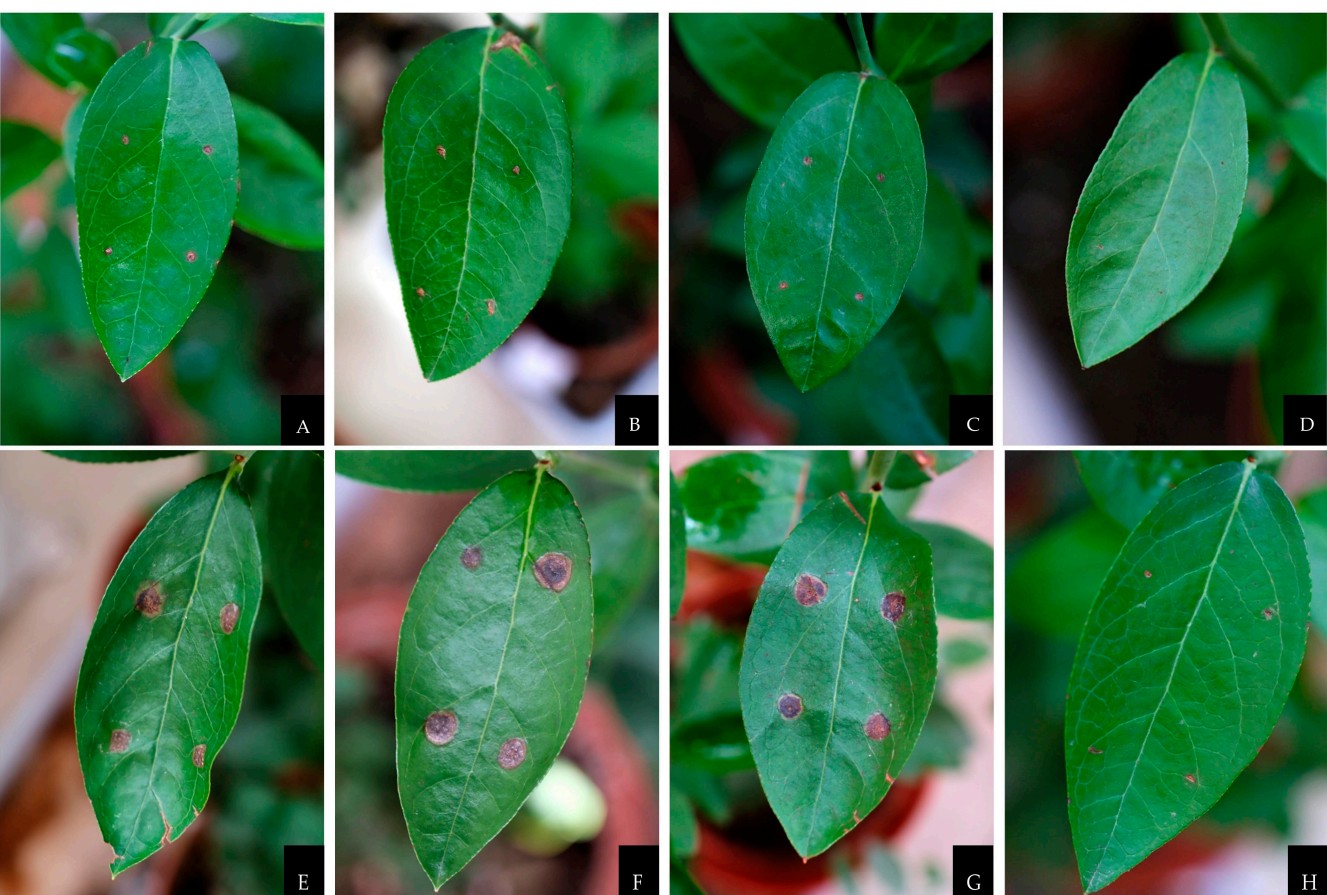

**Figure 2.** Pathogenicity of *Colletotrichum* isolates (NG5-1, NG5-4, and N2-1) on blueberries. (**A**–**C**) Symptoms on leaves of control plants inoculated with suspensions of NG5-1 (**A**), NG5-4 (**B**), and N2-1 (**C**), respectively, 3 days after inoculation. (**D**) Control leaves with sterile water treatment had no symptoms after 3 days of treatment. (**E**–**G**) Symptoms on leaves of control plants inoculated with suspensions of NG5-1 (**E**), NG5-4 (**F**), and N2-1 (**G**), respectively, 7 days after inoculation. (**H**) Control leaves with sterile water treatment had no symptoms after 7 days of treatment.

### 3.3. Multilocus Phylogenetic Analyses

Multilocus phylogenetic analyses (ITS, *CAL*, *ACT*, *TUB2*, *ApMat*, and *GAPDH*) results were obtained by maximum likelihood (ML) analysis in IQtree V 1.6.8 and Bayesian inference (BI) in MrBayes V 3.2.6. Four isolates (NG5-1, NG5-2, NG5-3, and NG5-4) clustered with *C. fructicola* (CGMCC3.17889). Meanwhile, two other isolates (N2-1, N2-2) were located in the same branch with *C. aenigma* (ICMP 18686) (Figure 3). Therefore,

NG5-1, NG5-2, NG5-3, and NG5-4 were identified as *C. fructicola*, and N2-1 and N2-2 were identified as *C. aenigma*. The ML and BI phylogenies were consistent in tree shape, and the final results were presented as ML trees. ITS, *CAL*, *ACT*, *TUB2*, *ApMat*, and *GAPDH* genes/regions of the five isolates were deposited in GenBank.

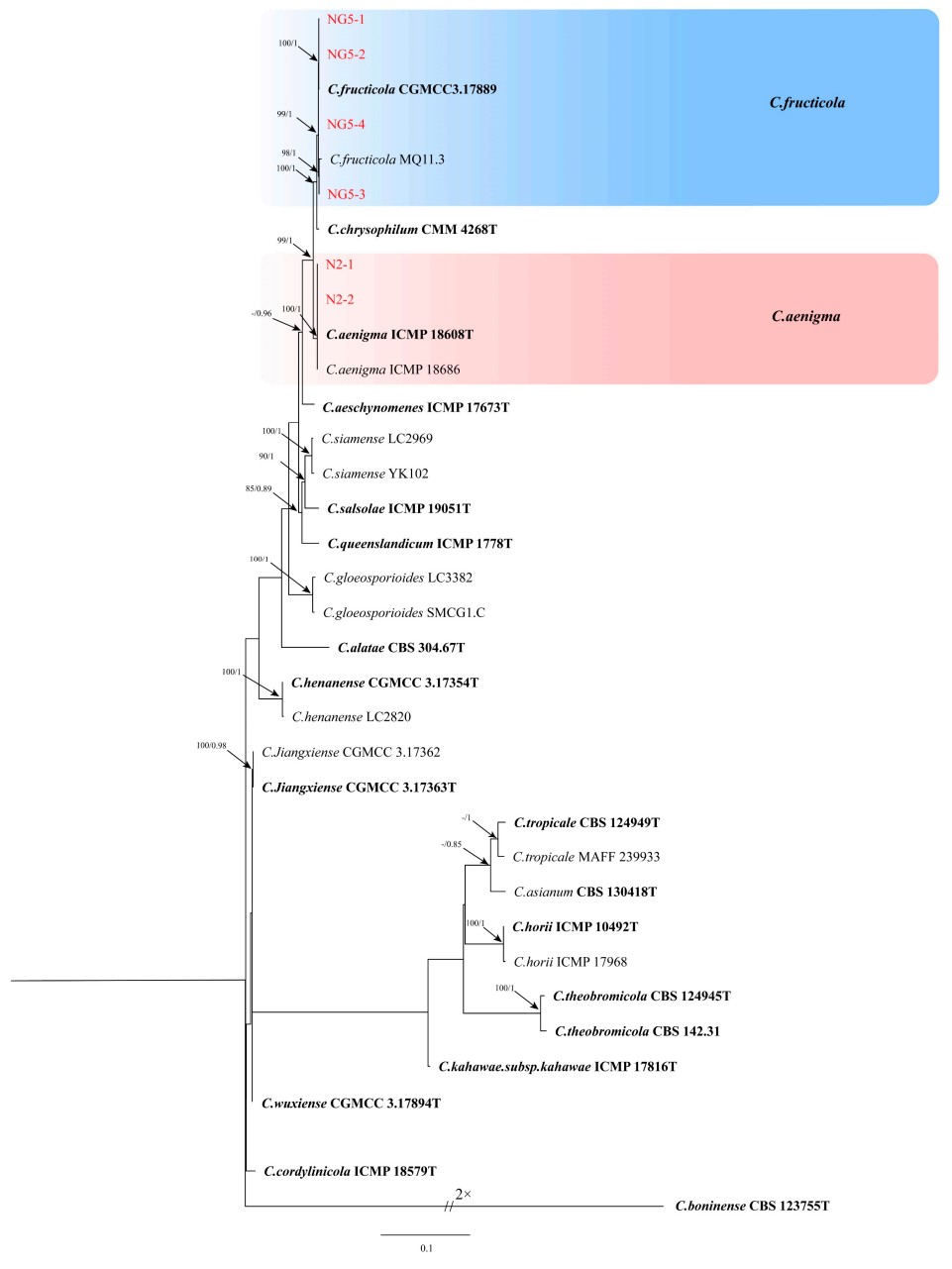

**Figure 3.** Phylogenetic relationships of N2-1, N2-2, NG5-1, NG5-2, NG5-3, and NG5-4 with related taxa were inferred using maximum likelihood and Bayesian posterior probability analyses based on the ITS, *ACT, TUB2, GAPDH, CAL*, and *ApMat* genes of *Colletotrichum* spp. Bootstrap support values (ML ≥ 80) and Bayesian posterior probability values (PP ≥ 0.80) are shown at the nodes (ML/PP), with *Colletotrichum boninense* (CBS 123755) as an outgroup. Bar = 0.1 substitutions per nucleotide position. Red font is the sequence of this study; bold indicates ex-types.

### 3.4. Morphology and Taxonomy

3.4.1. *Colletotrichum fructicola* Prihastuti, L. Cai and K.D. Hyde

**Culture characteristics**: On the PDA medium, the colonies exhibited round morphology, while the mycelium initially possessed a white, fluffy texture. Subsequently, after 3–5 days of cultivation, the central bulge turned olive-green or gray-green with white edges,

and the aerial mycelium was fluffy and densely cottony (Figure 4A). The colony's back was characterized by a black center with concentric rings having white edges (Figure 4B).

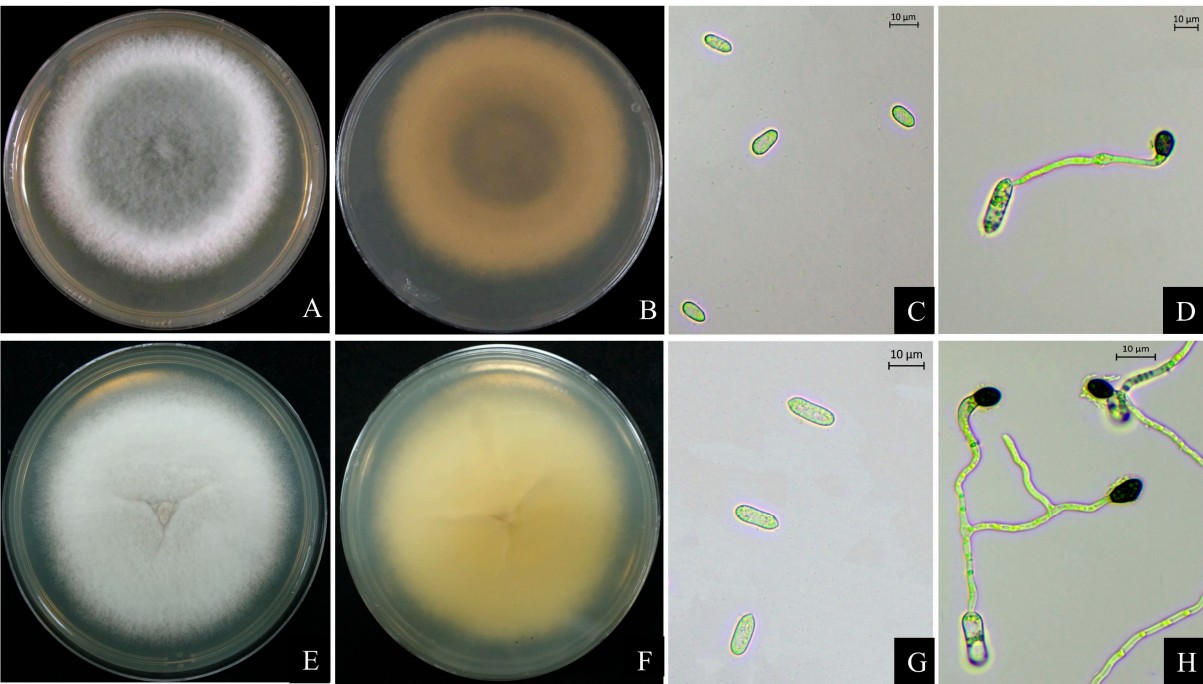

**Figure 4.** Morphological characterization of *Colletotrichum fructicola* (isolate NG5-4) and *C. aenigma* (isolate N2-1). (**A–D**). Isolate NG5-4: (**A**) front and (**B**) back views of 5-day-old culture on PDA medium. (**C,D**) Conidia, conidium, and appressorium, scale bars =10 μm. (**E–H**). Isolate N2-1: (**E**) front and (**F**) back views of 5-day-old culture on PDA medium. (**G,H**) Conidia, conidium, and appressorium, scale bars = 10 μm.

**Description**: Conidia were one-celled, hyaline, with smooth edges; spore shape was mostly long ellipsoid; both ends bluntly rounded or one end slightly pointed; and with size of (11.4–)13.2–15.3(–17.1) × (3.9–)5.0–6.4(–7.5) μm (mean ± SD = 13.7 ± 1.8 × 5.7 ± 1.1 μm) (Figure 4C). Appressoria were brownish in color, round or subrounded in shape, solitary, and (10.3–)11.6–14.0(–15.3) × (5.0–)5.9–7.3(–8.2) μm (mean ± SD = 11.5 ± 1.3 × 6.3 ± 0.9 μm) (Figure 4D).

**Notes**: Based on the results of phylogenetic analyses, four isolates (NG5-1, NG5-2, NG5-3, and NG5-4) clustered in the same clade as *C. fructicola* (Figure 3). The morphological characteristics of the four isolates were similar to those of *C. fructicola* [15]. NG5-1, NG5-2, NG5-3, and NG5-4 were identified as *C. fructicola* with the results of the morphological and phylogenetic analyses.

### 3.4.2. *Colletotrichum aenigma* B. Weir and P.R. Johnst

**Culture characteristics**: On the PDA medium, the colonies were sub-circular, with the whole front side white and a few centers in light gray. The colonies were flat, with a thick layer of fluffy aerial mycelium on top (Figure 4E). The back of the colony was light gray in the center and white overall (Figure 4F).

**Description**: Conidia were hyaline without septa, mostly cylindrical, with smooth and obtuse rounded ends; size (10.7–)12.7–18.8(–20.8) × (3.9–)4.6–6.5(–7.2) μm (mean ± SD = 14.2 ± 2.0 ×5.3 ± 0.7 μm) (Figure 4G). The appressoria were dark brown in color, ovoid or irregular in shape, and with a size of (5.9–)7.0–8.8(–9.9) × (3.7–)4.4–6.1(–6.8) μm (mean ± SD = 8.1 ± 1.1 × 5.2 ± 0.7 μm) (Figure 4H).

**Notes**: Based on the results of phylogenetic analyses, two isolates (N2-1, N2-2) were in the same clade as *C. aenigma* (Figure 3). The morphological characteristics of the two isolates were similar to *C. aenigma* [54]. N2-1 and N2-2 were identified as *C. aenigma* with

the results of the morphological and phylogenetic analyses. To our knowledge, this is the first report of *C. aenigma* causing anthracnose on blueberry leaves within China.

### 3.5. Biological Characteristics

3.5.1. Effects on Growth and Sporulation of C. aenigma at Different Temperatures, pH, and Light

*C. aenigma* was able to grow normally at temperatures of 5–35 °C. There were significant differences in mycelial growth and spore production at different temperatures ($p < 0.05$). With a colony diameter of 67.77 mm and sporulation of $15.83 \times 10^6$/mL at 25 °C and 61.47 mm and $20.57 \times 10^6$/mL at 30 °C, both were significantly higher than the other experimental groups. The strains grew most slowly at 5 °C and 35 °C. Therefore, the optimum temperature for mycelial growth was 25 °C, and the optimum temperature for sporulation of the strain was 30 °C (Figure 5A).

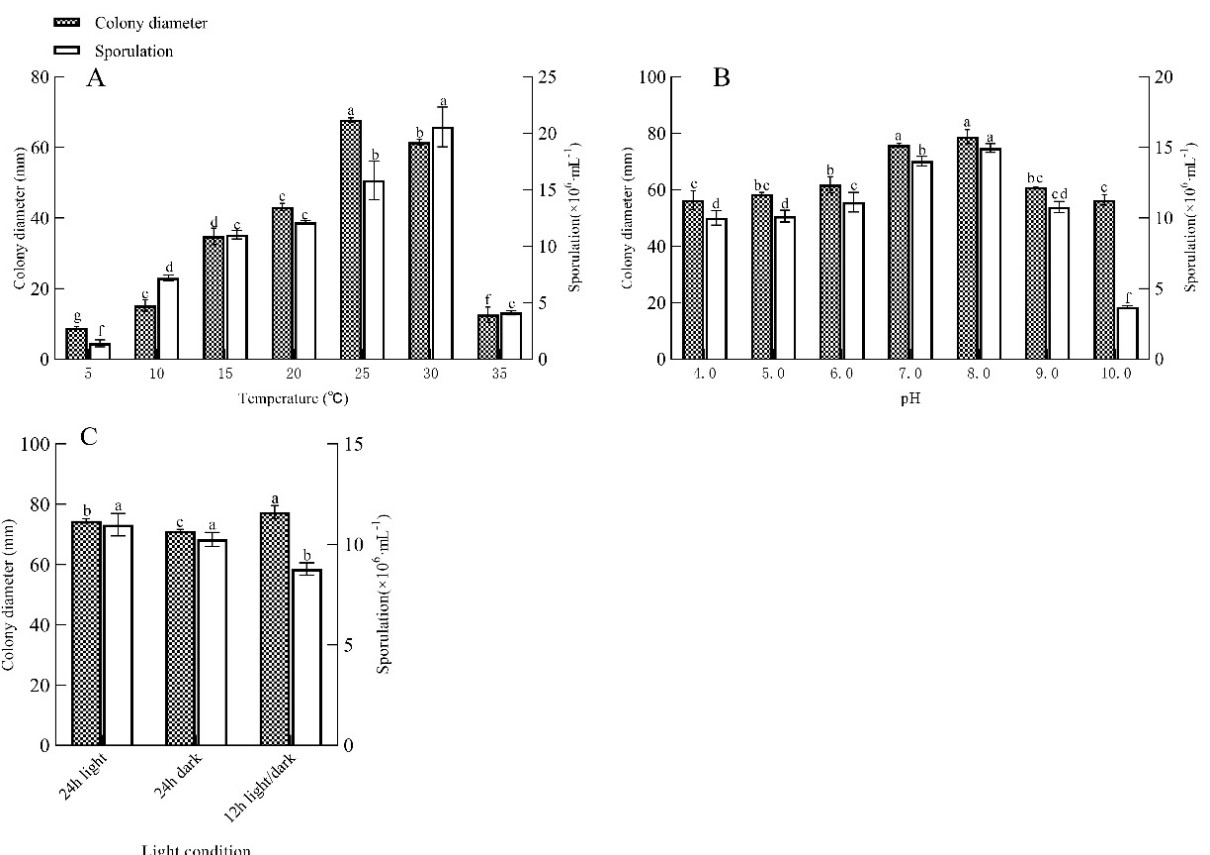

**Figure 5.** The impact of temperature, pH, and light on the mycelial growth and sporulation of *C. aenigma*. (**A**) Effect of temperature on the strains; (**B**) effect of pH on the strains; and (**C**) effect of light conditions on the strains. Lowercase letters on the histograms indicate different significances of variance. No significant difference between groups is denoted by the same letter, while a significant difference ($p < 0.05$) is indicated by different letters. It should be noted that we only performed multiple comparisons between bar graphs for the same indicators.

The isolate showed a wide range of pH adaptability, with normal growth and spore production in environments ranging from pH 4 to 10. The organism grew better in slightly acidic conditions than in alkaline environments. Significant differences in mycelial growth and sporulation were observed at different pH values ($p < 0.05$). At pH 7–8, colony growth and sporulation were significantly higher than those in other experimental groups, but the differences were not significant. At pH 8, the colony diameter was 78.79 mm, and sporulation was $14.95 \times 10^6$/mL. Therefore, the strain had an optimum pH of 7–8 for mycelial growth and sporulation (Figure 5B).

*C. aenigma* reached a colony diameter of 77.40 mm under a 12 h light/dark condition, which was significantly higher than the other two experimental treatments ($p < 0.05$). The strain's growth was slowest under the 24 h dark treatment, but the sporulation was the highest, reaching $10.98 \times 10^6$/mL. Therefore, the optimal condition for mycelial growth was 12 h light/dark, and for sporulation of the strain, 24 h dark (Figure 5C).

### 3.5.2. Effects of Different Culture Media and Carbon and Nitrogen Sources on Mycelial Growth and Spore Production

Significant differences were observed in colony diameter and sporulation of *C. aenigma* under different media ($p < 0.05$). Mycelium growth was dense on PDA with a maximum colony diameter of 71.80 mm, whereas on MM, the strain grew the slowest and mycelium was sparse, but the sporulation on MM was significantly higher than the other treatment groups, with a sporulation yield of $20.32 \times 10^6$/mL (Figure 6A). It demonstrates that PDA was the optimal medium for mycelial growth, while MM was the optimal medium for spore production.

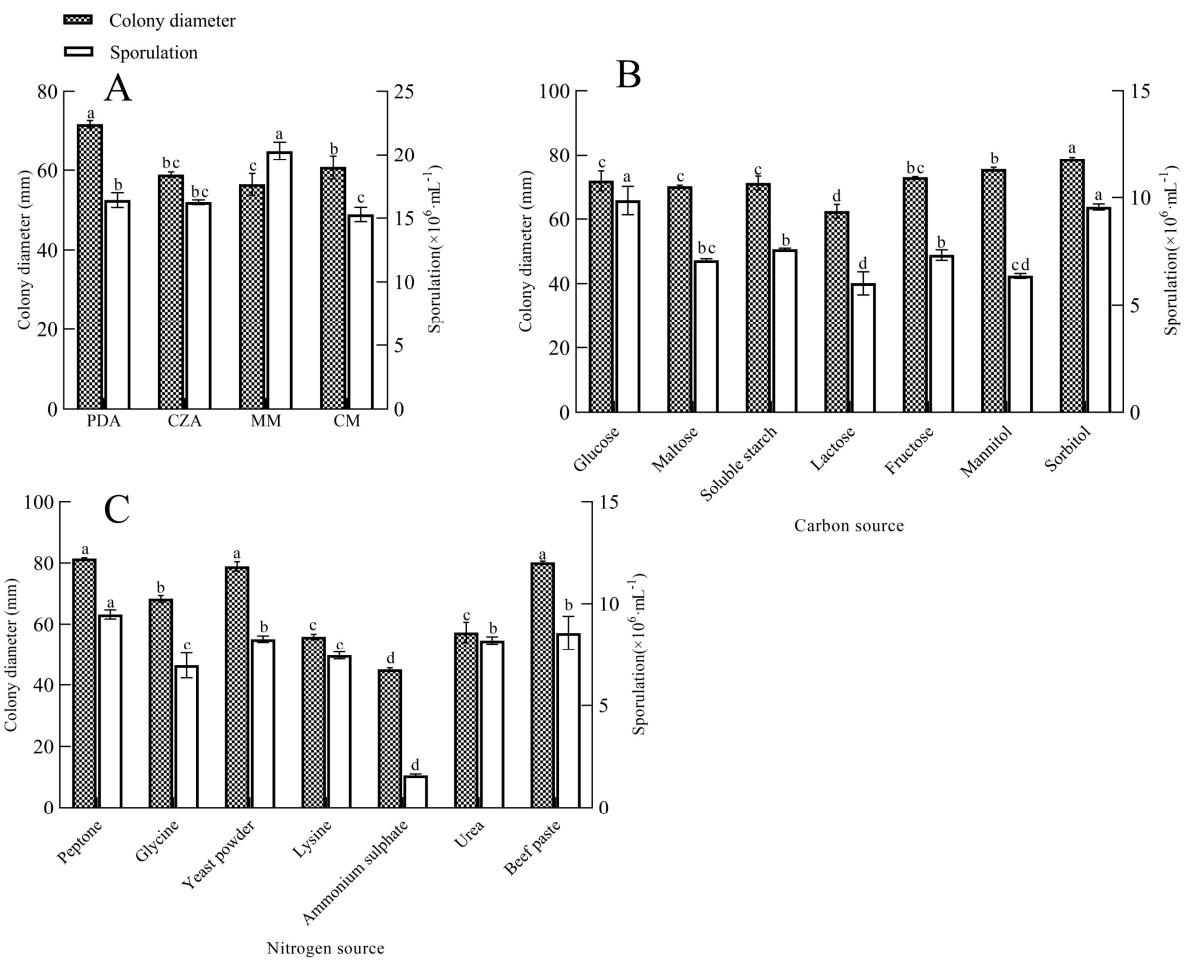

**Figure 6.** Effects of different media, carbon, and nitrogen sources on growth and spore formation of isolate N2−1. (**A**) Effects of different media on strains; (**B**) effects of different carbon sources on strains; and (**C**) effects of different nitrogen sources on strains. Lowercase letters on the histograms indicate different significances of variance. No significant difference between groups is denoted by the same letter, while a significant difference ($p < 0.05$) is indicated by different letters. It should be noted that we only performed multiple comparisons between bar graphs for the same indicators.

The utilization of different carbon sources by *C. aenigma* differed significantly ($p < 0.05$). It grew fastest in the medium with sorbic acid as the carbon source, which was significantly larger than other groups, and formed dense colonies in 7 days of cultivation, the diameter of

which reached 78.7 mm. In the medium with glucose as the carbon source, the sporulation was significantly greater than in the other groups, reaching $9.87 \times 10^6$/mL (Figure 6B). Therefore, the optimal carbon source for growth was sorbitol, and the optimal carbon source for spore production was glucose.

Significant variations were observed in the effects of various nitrogen sources on the mycelial growth and spore production of *C. aenigma*. When the nitrogen source was peptone, both colony diameter and spore production were markedly higher compared to other groups, reaching 81.47 mm in diameter and $9.48 \times 10^6$/mL in spore production ($p < 0.05$). Conversely, the strain exhibited the slowest growth and minimal spore production when the nitrogen source was thiamine (Figure 6C).

## 4. Discussion

Anthracnose fungi are ubiquitous pathogens in nature and represent significant phytopathogens of numerous economic crops, including *Fragaria × ananassa* Duch [55], *Vitis vinifera* L. [56], *Malus pumila* Mill [57], *Mangifera indica* L. [58], etc., resulting in negative impacts on plant growth and reduced yields, leading to substantial economic losses. Blueberry anthracnose is a significant disease in many countries, causing damage to blueberry production throughout the year by affecting leaves, branches, and fruits [14,31,59]. In previous reports, *C. gloeosporioides* and *C. acutatum* were confirmed as the main pathogens [60–62]. However, the morphological features of *Colletotrichum* spp. are influenced by environmental conditions, which can lead to close relatives being difficult to differentiate and classify accurately based on their morphological features and ITS sequences [16,63]. The advancement of molecular techniques has facilitated the successful distinction of many morphologically similar *Colletotrichum* species through the analysis of multiple specific genes or loci [20,39,64]. Therefore, the use of multilocus phylogenetic analysis combined with morphological characterization and pathogenicity analysis is currently the most effective method for identifying *Colletotrichum* spp. In this study, the pathogenicity of the representative strains was verified by Koch's postulates, and combined with morphological characteristics and ITS, *CAL*, *ACT*, *TUB2*, *ApMat*, and *GAPDH* multigene phylogenetic tree analyses, the pathogens that caused blueberry anthracnose in Nanjing were determined to be *C. fructicola* and *C. aenigma*, with *C. fructicola* as the dominant pathogen. This is the first report of blueberry anthracnose caused by *C. aenigma* in China, and there have been previous reports of *C. aenigma* infestation on a variety of plants such as strawberries [55], apples [57], and peppers [65].

The growth of *Colletotrichum* spp. is susceptible to environmental factors, including temperature, pH, light, carbon, and nitrogen sources. In our study, the optimal growth conditions for *C. aenigma* were 25–30 °C and pH 7–8, with the fastest mycelial growth and highest spore production at 25 °C. *C. aenigma* is more adaptable to different acidic and alkaline environments, with optimal growth at pH 7–8. The spore production under acidic conditions of pH 4–6 was greater than that under pH 9–10, indicating that acidic conditions are favorable to the spore production of the strain. This is generally consistent with other studies' findings [24,39,66–68]. The harvesting period of blueberry is from June to August every year, a period when daytime temperatures of around 20–30 °C and a pH of 6–7 in the host environment provide optimal growing conditions for *C. aenigma*. However, colonies hardly grow at temperatures between 5 °C and 35 °C, suggesting that anthracnose can be controlled by storing the fruit at low or suitably high temperatures after harvest.

Light levels are also a crucial factor in both growth and spore production for the pathogen. When exposed to a 12 h light/dark cycle, *C. aenigma* demonstrated the quickest mycelial growth. In the 24 h dark cycle, the fungus produced the maximum number of spores. It shows that light for a certain period of time favors the growth of the fungus, while conditions of complete darkness favor the spore production of the strain. Previous research has discovered that the 24 h light condition suppresses the spore production of various fungi, which is the same as the results of the present study [69–71]. *C. aenigma* was capable of growing and producing spores adequately on the media supplied for testing. Mycelium

growth was fastest on PDA, and spore production was highest on MM medium, indicating that the nutrients required for mycelial growth and spore production of the fungus varied and that the media selected had a direct effect on the growth of the pathogen [72]. *C. aenigma* grew normally in different carbon and nitrogen sources. However, the effects on colony growth and sporulation were inconsistent, with the most suitable carbon sources for colony growth and sporulation being sorbitol and glucose, respectively. The demonstrated ability of *C. gloeosporioides* to degrade starch, pectin, fat, and lignin during its penetration and colonization process has been established [73]. Consequently, it is plausible to infer that *C. aenigma*, as a constituent of the *C. gloeosporioides* species complex, possesses the capacity to decompose a diverse range of carbon sources. Moreover, *V. corymbosum* contains a plethora of amino acids, sugars, fats, dietary fibers, and other substances that promote the ideal host environment for the growth of *C. aenigma* [61]. The findings from this nitrogen source experiment indicate that peptone is most conducive to the growth and spore production of *C. aenigma*, while ammonium sulfate has an inhibitory effect on these processes. Hence, spraying ammonium sulfate-containing sprays directly onto the surface of blueberries may effectively control blueberry anthracnose.

## 5. Conclusions

In this study, the pathogens of blueberry anthracnose were identified as *C. fructicola* and *C. aenigma* by combining the results of pathogenicity validation, morphological characterization, and multilocus phylogenetic analyses. This finding further underscores the capability of multiple *Colletotrichum* species to infect a single host (*V. corymbosum*) and provides additional evidence for the host diversity of *C. aenigma*. This is the first report of *C. aenigma* as the causal agent of blueberry anthracnose in China. Further biological characterization showed that different factors, such as temperature, pH, light, and carbon and nitrogen sources, could affect the growth and spore production of *C. aenigma* to different degrees. Hence, during field management, develop appropriate control strategies to avoid conditions favorable to the growth of the pathogen and, therefore, control the disease. The findings of this study further enrich the pathogenesis of blueberry anthracnose and provide important information for future prevention and management strategies for this disease.

**Supplementary Materials:** The following supporting information can be downloaded at: https://www.mdpi.com/article/10.3390/f15010117/s1, Table S1. List of *Colletotrichum* spp. strains used for phylogenetic analysis.

**Author Contributions:** Conceptualization, W.-K.F.; Methodology, W.-K.F.; Formal analysis, W.-K.F.; Investigation, W.-K.F. and C.-H.W.; Data curation, C.-H.W.; Writing—original draft, W.-K.F.; Writing—review and editing, W.-K.F., Z.-X.C. and X.W.; Supervision, Y.-W.J. and D.-L.F.; Project administration, D.-L.F. and D.-L.F.; Funding acquisition, Y.-W.J. All authors have read and agreed to the published version of the manuscript.

**Funding:** This study was supported by a special fund from the Jiangsu Agricultural Industry Technology System (JATS[2022]510).

**Data Availability Statement:** All sequences generated in this study were submitted to GenBank.

**Acknowledgments:** The authors express sincere gratitude to those individuals who offered assistance and valuable insights throughout the course of this study.

**Conflicts of Interest:** The authors declare no conflict of interest.

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
