# Peer review of "Identifying the Biological Characteristics of Anthracnose Pathogens of Blueberries (Vaccinium corymbosum L.) in China"

_forests, doi:10.3390/f15010117_

Round 1

Reviewer 1 Report

Comments and Suggestions for Authors

Manuscript addresses a significant problem, which is anthracnose disease of Vaccinium corymbosum in China. Two main pathogens of this disease from the Colletotrichum genus have been found, including C. aenigma, which was demonstrated for the first time in such a role in China. There is no explanation why research was mainly limited to this one pathogen. The experiments that have been carried out are interesting and bring new aspects to science. However, the manuscript requires major revision. It contains numerous substantive and language errors. Even though there are six authors, none of them noticed that even the name of the blueberry Vaccinium corymbosum is misspelled (as corymbosun). The methodology is written unclearly and contains numerous gaps. Part of the methodology is only given in Results. The entire text must be processed and corrected. Only some errors or omissions are mentioned in the remarks.

Remarks

Line 2 -3 Title requires correction, remove analysis, use rather Pathogens – compare Conclusion line 385

Line 37 Vaccinium corymbosun L. - full name and author are required only when first used in the text, i.e. in line 36, later it should be V. corymosum, without author. It should be V. corymosum in whole manuscript and not V. corymbosun

Line 55 it should be Neofusicoccum vaccini

Line 66 it should be C. chlorophytii

Line 60 ‘as phytopathogens’ should be deleted, because the same thing is repeated in line 61

Line 90 more details should be provided, e.g. how many plants (or 1 leaf from one plant), how far the plants were from each other - what area did the samples represente, whether there were differences in the severity of symptoms on sampled leaves

Line 98 Pathogenicity test - the methodology is described in a unclear and incomplete way. Some aspects are yet to be written in Results

Line 102 The methodology is unclear - were all isolates obtained from leaves tested for pathogenicity, and how many leaves were inoculated in total?

Line 105  how long were they incubated and what was done with such leaves?

Line 106 to explain what these isolates were, why they were used for the test. The explanation is available only in Results, line 199

Line 113-114 unclear - Symptomatic leaves were reisolated ? fungi are reisolated - not leaves!

Line 114 The plants used in the experiment – please provide their characteristics, age or height, what pots did they grow in?

Line 120 There primers? (English)

Line 129 it should be conditions.

Line 130 Anglyses ??

Line 145 it is not clear what these strains are

Line 147-148 text completely unclear

Line 152 why only C. aenigma was analyzed

Line 160 it should be ‘….culture media mycelial plugs ..’

Line 164, line 171 text needs correction (English)

Line 180 it should be Fungi instead of Fungus

Line 181-185 you did not write in M&M that such analyses were performed, there is no test method

Line 187 what do you mean -85 fungi? - fungal colony

Line 209 text is unclear

Line 219 it should be Hyde (without dot)

Line 232, line 246 the text requires correction

Line 227, line 243 ‘m’ is usually given only once at the end of the dimensions

Line 278 please provide details if it was calculated in M&M (this is a problem when Colletotrichum in culture produces conidia mainly in sporodochia not on hyphae (such as Alternaria)

Line 288 it should be ‘sporulation (Figure 5B).’

Line 325 phytopathogen – in Singular?

Line 328 ‘Blueberry anthracnose is a significant pathogen’ – please note, anthracnose is disease not pathogen

Line 347 the optimal growth conditions for C. aenigma were 25-30°C and pH 4-10,?? Compare with line 283-284 and line 350??

Line 399- 405 Author contribution XX YY requires completion

Literature does not fully adhere to Forests Journal board. Some Latin names are in Italic, others are not.

Comments on the Quality of English Language

see remarks

Reviewer 2 Report

Comments and Suggestions for Authors

Dear authors: your work is an excellent contribution to the effort to control the spread of anthracnose on this crop. There are just a few suggestions to improve the text in general, as follows:

- All through the text - Vaccinium corymbosun – the correct name is Vaccinium corymbosum.

- Line 25 – Abstract: “C. aenigma exhibits optimal growth at a pH of 8” but in Line 288 – “an optimum pH of 7-8 for mycelial growth and sporulation”. Change to pH of 7-8. In line 350, again: “the optimal growth at pH 8”.

- Line 55 - Neofusicoccum Vaccinii – coorect is Neofusicoccum vaccinii.

- Line 66 - C. Chlorophyti – the correct is C. chlorophyti.

- In Figure 2 the leaf in “H” is not described. Correct to “Control leaves with sterile water treatment had no symptoms after 7 days of treatment (H).

- Figure 5 and 6 – It is somewhat difficult to read in the figure (not focused).

- Line 385 – include the scientific name of the blueberry in brackets or parentheses.

- at least a suggestion to include the scientific name in the title, as follows: “Identifying, and Analyzing the Biological Characteristics of 2 Anthracnose Pathogen of Blueberry (Vaccinium corymbosum L.) in China.”

Round 2

Reviewer 1 Report

Comments and Suggestions for Authors

After making corrections and supplementing the missing data, the manuscript should be accepted for printing in Forests. This work brings new valuable aspects to science. Only minor corrections are necessary because typographical errors or unclear passages (most notable examples presented in Remarks) appear.

Remarks

Line 69 it should be [27].

Line 93 this sentence needs revision

Line 158 what do you mean "characterized by Colletotrichum"?

Lines 183-187 this text is exactly repeated in Lines 195-199. This requires explanation

Line 214 ‘isolates were isolated’ – may be colonies were isolated

Line 254 consider revising this sentence (no space)

Line 381 it should be pH 7-8

Comments on the Quality of English Language

see Remarks
